# Cost-Effectiveness Analysis of Perioperative Oral Management after Cancer Surgery and an Examination of the Reduction in Medical Costs Thereafter: A Multicenter Study

**DOI:** 10.3390/ijerph18147453

**Published:** 2021-07-13

**Authors:** Hideki Sekiya, Yasuhiro Kurasawa, Yutaka Maruoka, Hitoshi Mukohyama, Akihide Negishi, Shiro Shigematsu, Junpei Sugizaki, Masaru Ohashi, Shiro Hasegawa, Yutaka Kobayashi, Masayuki Ueno, Yukihiro Michiwaki

**Affiliations:** 1Department of Oral Surgery, School of Medicine, Toho University, Tokyo 143-8541, Japan; yukirom@musashino.jrc.or.jp; 2Department of Oral & Maxillofacial Surgery, Tokyo Medical and Dental University, Tokyo 113-0034, Japan; yasukura1408@gmail.com; 3Department of Oral & Maxillofacial Surgery, Center Hospital of the National Center for Global Health and Medicine, Tokyo 162-8655, Japan; ymaruoka@hosp.ncgm.go.jp; 4Department of Oral & Maxillofacial Surgery, Yokohama City Minato Redcross Hospital, Yokohama 231-8682, Japan; mukouyama.osur@yokohama.jrc.or.jp; 5Department of Oral & Maxillofacial Surgery, National Hospital Organization Yokohama Medical Center, Yokohama 245-8575, Japan; negishi.akihide.gf@mail.hosp.go.jp; 6Department of Oral & Maxillofacial Surgery, Tokyo Metropolitan Tama Medical Center, Tokyo 183-8524, Japan; shirou_shigematsu@tmhp.jp; 7Department of Dentistry, Toranomon Hospital, Tokyo 105-8470, Japan; j-sugizaki@toranomon.gr.jp; 8Department of Oral & Maxillofacial Surgery, Japan Community Health Care Organization Tokyo Takanawa Hospital, Tokyo 108-8606, Japan; ohashi@dent.showa-u.ac.jp; 9Department of Dentistry & Oral Surgery, Tokyo Metropolitan Health and Hospitals Corporation, Ebara Hospital, Tokyo 145-0065, Japan; shirou_hasegawa@tokyo-hmt.jp; 10Department of Oral & Maxillofacial Surgery, Tokyo Metropolitan Hiroo General Hospital, Tokyo 150-0013, Japan; yutaka_kobayashi@tmhp.jp; 11Department of Health Science, Saitama Prefectural University, Saitama 343-8540, Japan; masayuki-ueno@spu.ac.jp; 12Department of Oral & Maxillofacial Surgery, Musashino Redcross Hospital, Tokyo 180-8610, Japan

**Keywords:** perioperative oral management, cancer surgery, incidence of postoperative pneumonia, incremental cost-effectiveness ratio, effect of reducing healthcare cost, oral care

## Abstract

In April 2012, perioperative oral management (POM) was approved for inclusion in the national health insurance system of Japan to prevent the occurrence of pneumonia, a major complication in cancer patients. The subsequent decrease in the incidence of postoperative pneumonia indicated the prophylactic effect of POM. The constant increase in health expenditure necessitates a cost-effectiveness analysis. In addition, the effect of reducing healthcare costs owing to health technologies must be evaluated. In the present multi-institutional study, the cost-effectiveness analysis of POM was conducted by comparing the incidence of postoperative pneumonia and the healthcare costs between patients who received surgery for malignant tumors before (*n* = 11,886) and after (*n* = 13,668) the introduction of POM. Additionally, the effect of reducing healthcare costs was evaluated. Reductions in the number of patients who developed pneumonia, duration of hospitalization, and number of deaths were observed after the introduction of POM. The incremental cost-effectiveness ratio was 111,927 yen, hence the prevention of postoperative pneumonia needs 111,927 yen per patient in healthcare costs. Consequently, a maximum reduction of 250,368,129 yen in healthcare costs was observed between the incremental costs for pneumonia treatment and the cost of POM. These findings indicate that improvements in cost-effectiveness can be expected in the future through the development of procedure and system for POM.

## 1. Introduction

The oral cavity is anatomically connected to the lungs through the pharynx, larynx, trachea, and bronchi. Therefore, contaminants within the oral cavity have the potential to cause pneumonia. However, the risk of pneumonia in the oral cavity is compensated by the process of swallowing; thus, a deterioration in the function of swallowing increases the risk of pneumonia, particularly, aspiration pneumonia [1,2]. Teramoto et al. [3] investigated the occurrence of pneumonia in healthcare facilities, where the indoor environment is maintained relatively clean, and reported that more than 80% of elderly patients with pneumonia presented with aspiration pneumonia. The risk of pneumonia increases after surgical treatment. In a study comprising approximately 2.5 million cancer patients aged 18 years or older who received surgery, the incidence of postoperative pneumonia was reported to be 3.5% [4]. Thus, the prevention of pneumonia in hospitalized patients, especially those with malignant tumors, is an important issue in Japan.

Consequently, the “perioperative oral management fee” was listed in the fee schedule for insured medical treatments (universal health insurance system in Japan) in April 2012 for the prevention of pneumonia in patients with malignant tumors. Since then, various hospitals affiliated with medical schools and municipal hospitals initiated the perioperative management of oral hygiene and function at the time of surgery, under general anesthesia, through collaborations between the medical and dental teams.

Several studies have demonstrated the preventive effects of POM on pneumonia [5,6]. Soutome et al. [7] reported a reduction (odds ratio, 0.42) in the incidence of postoperative pneumonia in patients with esophageal cancer who received oral management (intervention group; *n* = 234) when compared to those who did not receive any intervention (*n* = 149). In another large-scale study comprising the healthcare reimbursement database in Japan, Ishimaru et al. [5] investigated approximately 509,179 patients with head and neck, esophageal, gastric, colon, lung, and liver cancers and found that patients who received POM (*n* = 81,632) showed a significantly lower incidence of postoperative pneumonia and 30-day postoperative mortality compared to those who did not receive POM (*n* = 427,547). Thus, the prophylactic effects of POM have been demonstrated in both a small and detailed analysis and a large-scale statistical analysis.

On the other hand, a cost-effectiveness analysis that includes the perspective of cost management, in addition to effectiveness and safety, has been increasingly adopted. In 2012, the Special Committee on Cost-Effectiveness was established within the Central Social Insurance Medical Council, and a cost-effectiveness analysis has been implemented on a trial basis since 2016 [8].

In addition to the new technologies and drugs, existing technologies concerning the national health insurance price list and drugs for reevaluation are evaluated in the cost-effectiveness analysis. The effectiveness is determined by evaluating the recurrence, survival rate, and quality-adjusted life-year (QALY), whereas cost includes both direct and indirect costs [9]. The cost evaluation method varies depending on the perspective, but the term “cost” in cost-effectiveness analysis is usually defined as direct costs incurred by the public healthcare system, instead of the healthcare providers or patients. The cost and effectiveness are calculated separately to obtain the incremental cost-effectiveness ratio (ICER), which compares the benefits of introducing a health technology and the associated increase in cost. When the new technology is costlier than and as effective as or less effective than the technology being compared to, it is considered as dominated. If the new technology is less costly and as effective as or more effective, it is considered as dominant. If it is costlier and more effective than the technology being compared to, the decision is made by setting a threshold for the ICER. The technology that is used for comparison may include no treatment or watchful waiting [9].

POM is not considered as a new technology but rather a new treatment system, and to the best of our knowledge, no health economics study or cost-effectiveness analysis on POM has been reported in the literature so far. The cost-effectiveness analysis of new treatment systems has social significance and is required for the development and improvement of new technologies.

In the present study, the incidence of postoperative pneumonia (effectiveness) and the treatment costs were compared before and after the introduction of POM in approximately 25,000 patients who received surgery for malignant tumors. The patients were selected from approximately 370,000 patients whose data were collected as part of a multi-institutional study on postoperative pneumonia [6].

This data is based on the same database as the paper by Kurasawa et al. published in 2020 [6]. Kurasawa et al. belongs to the same study group and is in charge of the statistics. For the data of about 25,000 cancer patients who underwent surgery, the incidence of pneumonia before and after the introduction of insurance was discussed using multivariate analysis considering confounding factors, and it was concluded that the introduction of insurance significantly reduced the incidence of pneumonia [6].

In addition, the protocol for the technique of perioperative oral management has not been standardized at the eight institutions. However, according to Sekiya et al. [10], this management method is positioned as a conventional system in the Japanese classification of perioperative oral management systems. In other words, this is a system in which professional oral management is performed only when the surgeon in charge requests oral management before the operation. As a result, in every facility, under the plan of the dentist or oral surgeon, the dental hygienist removes tartar and teaches tongue and tooth plaque removal, and the dentist extracts unstable or infected teeth.

## 2. Subjects and Methods

### 2.1. Subjects

Eight regional core hospitals in the metropolitan area, which were equivalent in terms of the healthcare resources, number of patients, and target diseases, were included in this study. All the institutions used the reimbursement system of bundled payment adopting the Diagnosis Procedure Combination (DPC) for their patients. All electronic data, including the patients’ clinical information and the records of the clinical procedures, were included in the DPC data.

In terms of the experience of surgery for malignant tumors, patients who presented with “97” (other surgery), “99” (no surgery), and unknown (such as xx) as the ninth and tenth digits of their DPC codes were assigned to the no surgery group, while the remaining patients were assigned to the group that received surgery for malignant tumors.

The period of analysis extended from two years before the introduction of POM in April 2012 (1 April 2010 to 31 March 2012) to two years after its introduction (1 April 2012 to 31 March 2014). The total number of hospitalized patients in all eight institutions during the four years was 367,666, and 81,859 of them had malignant tumors.

The exclusion criteria were patients with malignant tumors who were hospitalized due to pneumonia, had pneumonia as a concurrent disease at the time of admission, and had not received surgery. As a result, the total number of patients with malignant tumors who received surgery was 25,554, including 11,886 and 13,668 patients who received the surgery two years before and after the introduction of POM, respectively (Figure 1).

Approvals for this multi-institutional study were obtained from the respective institutional review boards.

### 2.2. Survey Items

Basic information regarding the month of discharge, department of discharge, age, sex, length of hospital stay and additional information about the disease structure (reason for hospitalization, comorbidity on admission, diseases developed since admission, DPC diseases, and DPC code) were extracted from the DPC data of the hospitalized patients and examined retrospectively. The costs were defined as fees based on fee-for-service as recorded in the DPC data.

### 2.3. Cost-Effectiveness Analysis of POM

Cost-minimization, cost-effectiveness, cost-utility, and cost-benefit analyses were used to analyze the cost-effectiveness of POM; the main difference among them was the way the outcome was treated. The cost-minimization analysis compares only the costs, assuming that the outcomes are equivalent, while the cost-effectiveness analysis uses outcomes other than QALY, such as life year and incidence of events. The cost-utility analysis uses QALY as the outcome, whereas cost-benefit analysis evaluates outcomes by expressing them in monetary terms.

The outcome in this study was the incidence of postoperative pneumonia without considering the quality of life; hence, the cost-effectiveness analysis was used. The costs were expressed as the reimbursement fees under the national health insurance scheme (in Japanese yen).

### 2.4. Incremental Cost-Effectiveness Ratio

The ICER is an evaluation index for cost-effectiveness analysis and can be calculated using the following formula: (1)$$ICER=\frac{IC}{IE}=\frac{C_{A}-C_{B}}{E_{A}-E_{B}}$$
where *IC* is the incremental cost, *IE* is the incremental effectiveness, *C_A_* is the cost of new therapy A, *C_B_* is the cost of control therapy B, *E_A_* is the effectiveness of new therapy A, and *E_B_* is the effectiveness of control therapy B.

The costs of the drug and the surgery as well as the tests were different before and after the introduction of POM (2010 to 2014) due to the revision of the fee schedule every two years. Therefore, the ICER was calculated based on the fee schedule after the introduction of POM (after April 2012).

The cost calculations were based on three assumptions. The first assumption was that, during the period prior to the introduction of POM (April 2010 to March 2012), 0% of the total patients in the eight participating hospitals received POM because the concept of POM for cancer patients did not exist at that time, although it was performed before cardiovascular and transplantation surgery. The analyses were limited to cancer surgery because of the initial introduction of the patients into the Japanese social healthcare system and the lack of perioperative oral management for cancer patients prior to the addition to the fee schedule, which simplified the statistical analyses. The second assumption was that the rate of patients who received POM among the eight hospitals was an average of 12.3% following its introduction (April 2012 to March 2014). Since the medical DPC data and dental data are sent to the data center in an unlinked and anonymized form, the dental intervention rate is discussed as unknown in the paper by Kurasawa et al. [6]. In this study, we calculated a dental intervention rate of 12.3% by manual data matching at the Yokohama City Minato Red Cross Hospital, where the data remained in the hospital in a linkable form. This was used as the second assumption. The third assumption was that, if no patient received POM after its introduction in the national health insurance scheme, the incidence rate of postoperative pneumonia would be same as that before the introduction of POM. The 25,554 subjects in this study were divided into two groups: before and after the introduction of POM. The third assumption was cleared by comparing the basic patient information, including age, sex, and comorbidities, which are considered as the background factors of pneumonia, between the two groups (Table 1 [6]). No significant differences were noted between the two groups. However, introducing of insurance system is not the only factor involved in the reduction of pneumonia; Kurasawa et al. conducted a multivariate logistic analysis using the same database to investigate the involvement of confounding factors [6]. Other confounders included: older than 70 years, male, brain tumor, esophageal cancer, gastric cancer, and lung cancer surgery (*p* < 0.01). Of these, in the post-implementation group, the mean age decreased and the proportion of males decreased, but the proportions accounted for by brain tumor, esophageal cancer, gastric cancer, and lung cancer surgery decreased, and it was unlikely that these factors had any effect on reducing the incidence of pneumonia.

Based on the third assumption, the expected number of patients who would develop pneumonia if they did not receive POM was determined as follows: the total number of hospitalized patients after the introduction of POM (13,668) was multiplied by the incidence of postoperative pneumonia and before the introduction of POM (2.01%) = 275 patients. Thus, the effectiveness of POM was determined using the following formula: 

*IE* (incremental effectiveness) = *E_A_* (actual number of patients with pneumonia after POM introduction: 114) − *E_B_* (expected number of patients who develop pneumonia before POM: 275) = −161 patients (estimated decrease in the number of patients with postoperative pneumonia).

The incremental cost was calculated based on the first and second assumptions, wherein 12.3% (1681 patients) of all patients eligible for POM (13,668 patients) following its introduction received the procedure. The total amount for POM fees (18,020,320 yen) was obtained by multiplying 1072 points (10,720 yen) per patient with the number of patients who received POM (1681 patients). The total number of points per patient was obtained by summing up the following items: initial dental consultation fee (282 points); perioperative oral management set-up fee (300 points); preoperative oral management fee I (190 points); and postoperative oral management fee II (300 points). As the cost prior to the introduction of POM was 0 yen, the incremental cost would be 18,020,320 yen. Furthermore, the points for POM were calculated as per the 2012 revision of the schedule for medical fees under the national health insurance scheme.

### 2.5. Method Used to Calculate the Decrease in Treatment Costs for Postoperative Pneumonia

In general, the duration of hospitalization tends to be short if the patient does not develop any complication. In such cases, the hospitalization fees include the fees for the treatment of the underlying disease (cancer-related treatment fees) and the comorbidities (Treatment Cost A). If the patient develops any complications following surgery, treatment for the complications is provided in parallel with the POM-related treatment. The patient is discharged after recovering from the complication. The period of hospitalization is prolonged if the patient develops complications, and the treatment costs for the complications are added to Treatment Cost A in such cases (Figure 2). However, it is not easy to segregate each treatment cost item; moreover, it is difficult to extrapolate the treatment cost for pneumonia only.

Thus, in this study, the treatment cost for pneumonia was defined as the difference between the total treatment cost and Treatment Cost A in a patient with postoperative pneumonia. Thereby, Treatment Cost A was deemed to be the same as the average cost borne by a patient without postoperative pneumonia.

## 3. Results

### 3.1. Comparison of Patients before and after the Introduction of POM

The potential factors related to postoperative pneumonia among cancer patients who received surgery included sex, age, site of cancer, and comorbidities, such as cerebral infarction, intracranial hemorrhage, and diabetes. The comparison of each factor before and after POM introduction is presented in Table 1 [6]. The total number of patients was increased by 1802 after the introduction of POM. The female-to-male ratios were 1.16 and 0.93 before and after POM introduction, respectively, indicating a slight reduction in the percentage of female patients. The mean ages of the patients before and after the introduction of POM were 61.0 ± 13.7 years and 64.7 ± 13.9 years, respectively. Thus, >10% difference in the number of target patients was observed between the groups before and after the introduction of POM. Therefore, a comparison was made by calculating the expected value after POM introduction based on the actual value before introduction and comparing it with the actual value observed after the introduction of POM.

The numbers of patients before and after the introduction of POM based on cancer site were 535 and 526 for brain tumors, 220 and 227 for head and neck cancers, 529 and 553 for esophageal cancers, 2060 and 2177 for gastric cancers, 1493 and 1657 for colon cancers, and 1002 and 990 for lung cancers, respectively. There is no significant difference between two groups before and after the introduction of POM.

### 3.2. Relationship between the Length of Hospital Stay and Postoperative Pneumonia

A significant difference in the length of hospital stay in patients with (29.8 days) and without (17.5 days) postoperative pneumonia before the introduction of POM was observed (Mann-Whitney’s *u* test, *p* < 0.001; Table 2). Likewise, the durations of hospital stay in patients with and without pneumonia were significantly different after the introduction of POM (45.4 days vs. 16.2 days, respectively; *p* < 0.001) (Figure 2).

### 3.3. Relationship between Treatment Cost and Postoperative Pneumonia

Hospitals are required to submit the treatment costs based on the fee-for-service model, despite the adoption of the bundled payment method by the DPC. Before the introduction of POM, significant differences in treatment costs in patients with and without postoperative pneumonia were observed (1,772,632 yen vs. 1,274,746 yen, respectively; *p* = 0.013; Table 3). Similarly, a significant difference between the two groups was observed after the introduction of POM (2,962,771 yen vs. 1,295,762 yen, respectively; *p* < 0.001; Table 3).

### 3.4. Relationship between Hospital Mortality and Postoperative Pneumonia

Before the introduction of POM, hospital mortality in patients with pneumonia (4.18%) was significantly higher than that in those without pneumonia (0.61%; *p* < 0.001; Table 4); similar results were observed after the introduction of POM (6.14% vs. 0.41%; *p* < 0.001).

### 3.5. Incidence of Postoperative Pneumonia

The number of patients with postoperative pneumonia was reduced by half from 239 patients (2.01%) before POM introduction to 114 patients (0.83%) after the introduction (Chi-square test; *p* < 0.001; Table 5).

### 3.6. Analysis of the Cost-Effectiveness of POM

Based on the aforementioned third assumption, the incidence rate of pneumonia in eligible patients (13,668 patients) following the introduction of POM was 2.01%, which was the same as that before the introduction of POM; thus, 275 patients were expected to contract pneumonia. However, in reality, 114 patients developed pneumonia, indicating a reduction in the number of patients who developed postoperative pneumonia. If all 275 patients who were hypothesized to develop pneumonia had received POM, its occurrence might have been prevented in a maximum of 161 patients.

Therefore, the incremental effectiveness would be the decrease in the number of patients with postoperative pneumonia (*n* = 161), and the incremental cost would be the fees for the POM (18,020,320 yen). The resultant ICER would be 111,927 yen. In other words, 111,927 yen was required to reduce the number of patients with postoperative pneumonia by one.

### 3.7. Effect on Reduction in Medical Cost

The average treatment cost (Treatment Cost A) for patients without postoperative pneumonia was 1,295,762 yen (±1,007,162) per patient and that for those with postoperative pneumonia was 2,962,771 yen (±1,964,419). Thus, the treatment cost for postoperative pneumonia was calculated as 1,667,009 yen per patient, which is the difference between the two average costs.

It is assumed that the procedure prevented the development of pneumonia in 161 patients following the introduction of POM. Hence, the total treatment cost would be 268,388,449 yen (1,667,009 yen × 161). As the total cost for POM was 18,020,320 yen, it was estimated that a total of 250,368,129 yen (approximately 250 million yen) towards medical expenses had been saved for 13,668 patients.

## 4. Discussion

### 4.1. Relationship between Postoperative Pneumonia and POM

Pneumonia can be classified into four types as follows: aspiration, ventilator-associated, bacterial, and acute. Aspiration and ventilator-associated pneumonia are thought to be involved with the contamination of the oral cavity [11], whereas correlations between oral cavity contamination and the two other types of pneumonia are unknown. Unlike community-acquired pneumonia, the majority of the cases of hospital-acquired pneumonia are considered to be aspiration pneumonia [3]. A more inclusive diagnosis, such as bacterial pneumonia and acute pneumonia, may be reached in some cases due to difficulties in reaching a definitive diagnosis. In the present study, pneumonia that newly developed after hospitalization (postoperative pneumonia) was considered to be associated with oral cavity contamination and could be prevented by POM.

### 4.2. Significance of a Reduction in the Number of Patients with Postoperative Pneumonia

According to the vital statistics in 2014 [12], pneumonia is the third leading cause of death in Japan, and older patients account for 90% of these deaths. Aspiration pneumonia caused by dysphagia accounts for at least 70% of the pneumonia cases in older patients [13]. Furthermore, older patients account for at least 70% of the hospital-acquired pneumonia cases [3]. Therefore, measures to prevent postoperative pneumonia are crucial in the current super-aging society.

In addition to the prevention of nutritional disorders and local oral infections, the purpose of POM is to prevent the occurrence of pneumonia and sepsis in the more distant sites. In particular, the preventive effects of POM on pneumonia are highly expected and demonstrated in the literature [2,5,6,7]. In the present study, the incidence of postoperative pneumonia was 2.01% before the introduction of POM, which decreased to 0.83% after the introduction. This result is lower than previously reported for pneumonia incidence.

Using the same target data as in this study, Kurasawa et al. [3] conducted a multivariate analysis to investigate the relationship with confounding factors, and showed the effectiveness of the introduction of oral management in preventing pneumonia after cancer surgery.

Furthermore, the results of the present study indicated that the length of hospital stay was prolonged, treatment cost was increased, and hospital mortality was higher in patients with postoperative pneumonia when compared to those without pneumonia. The ratio of extension in the duration of hospital stay in patients with pneumonia compared to those without pneumonia was approximately 1.7 before POM introduction, which increased to 2.8 approximately two years after POM introduction (Table 2). The ratio of increase in treatment costs in patients with pneumonia when compared to those without pneumonia was 1.3 before POM introduction and 2.2 after its introduction (Table 3). Moreover, the mortality rate in patients with pneumonia compared to those without pneumonia was 6.8 before POM introduction, which had increased to 14.9 after POM introduction (Table 4). These changes occurred within a relatively short period of approximately four years, thereby indicating its gravity in terms of healthcare and medical economics in Japan. It is difficult to consider this rapid change based on annual trends alone, and there may be an effect of the introduction of POM. Even in the group of patients who did not have postoperative pneumonia, it should be assumed that the introduction of POM had a cost-saving effect. This suggests the need for introduction of POM to hospitals without dental systems, and the need for a sustainable system of POM.

### 4.3. Cost-Effectiveness Analysis of POM: Evaluation of Incremental Effectiveness Calculation

In this study, incremental effectiveness was defined as a decrease in the incidence rate of postoperative pneumonia in cancer patients. The expected values based on the three aforementioned assumptions were obtained.

The first assumption was that no eligible patient underwent POM or any similar procedure during the period prior to the introduction of POM (April 2010 to March 2012). During this period, certain individual arrangements had been made; for example, a patient could receive preoperative treatment at a dental clinic or/and oral surgery department and have the source of infection removed, if requested by a cardiovascular surgeon. However, no procedure equivalent to POM was performed before cancer surgery.

Hence, the introduction of POM to the national health insurance scheme required hospitals to develop a new system that would allow them to perform the procedure effectively. Since patients eligible for POM were already receiving medical treatment, they were unlikely to voluntarily visit a dentist for POM and would need to be referred to a dentist by a physician, when necessary. The physicians and nurses were informed about the significance of POM, and that performing this procedure would not burden the health professionals or the patients. In addition, the physicians and nurses were provided with information about the standard explanations that must be provided to cancer patients who are not convinced about the benefit of POM, the ideal time for a patient to receive POM, the recommended procedures, and the anticipated financial burden on the patient. In the present study, less than 1% of patients received preoperative dental treatment before POM introduction in or before 2012.

For the second assumption, we conducted a survey at the participating hospital after POM was introduced. The results of the survey indicated that 12.3% of the total patients received preoperative dental treatment. Although it cannot be exhaustively concluded that dental treatment is the same as POM, we assumed that the patients had received POM. Ishimaru et al. [5] evaluated the Japanese national medical fee database and examined the data of approximately 510,000 patients who underwent cancer surgery from May 2012 to December 2015, after the introduction of POM and found that 16.0% patients had received preoperative dental treatment. Therefore, given that the dental intervention rate in this study was still low (12.3%) in 2011–2012 and 16% [5] by 2015, it is possible that the intervention rate will increase further and as a result the incidence of pneumonia will be lower than in this study. A further reduction in health care costs could be expected. Dental interventions increase health care costs by 3.7%, but the incremental cost is negligible.

The third assumption was that, if no eligible patient received POM after its introduction to the national health insurance scheme, the incidence rate of postoperative pneumonia (2.01%) would remain the same as that before the introduction of POM. Although cases from two different periods were compared, they were all considered equivalent through a head-to-head comparison of the aggregated data without adjusting the background factors. This may be considered as a limitation of this study. A preceding study by Ishimaru et al. [5] used data from the Japanese Nationwide Administrative Claims Database and compared the incidence of pneumonia between approximately 80,000 patients who received POM and approximately 420,000 who did not receive POM during the same period. As there was a five-fold difference in the number of patients between these two groups, background factors, such as sex, age, diagnosis, prescription, and medical procedures, were adjusted by propensity score matching analysis. The researchers then weighted the data by the inverse probability of treatment weighting method before comparing the groups.

From a clinical perspective, the onset of postoperative pneumonia in cancer patients involves varying factors, in addition to those suggested by Ishimaru et al. [5], such as the type and stage of cancer (e.g., the TNM staging system and stage classification systems), the preoperative respiratory function and body mass index; the degree of surgical invasiveness (e.g., surgical techniques, surgery duration, blood loss amount), postoperative respiratory management, and postoperative complications (e.g., circulatory disorders, renal dysfunction, hepatic dysfunction). Owing to the fact that different scales are adopted for these factors, a precise analysis would require references to preceding studies that have systematically elucidated inter-factor relationships (confounding factors) and those that weigh each factor by setting postoperative pneumonia in cancer patients as an objective variable. Additionally, the number of sample cases for obtaining reliable results through logistic analysis is required at least ten times as many as the number of objective variables [13]. This requirement was met in the current study where the number of subjects was 11,886 and 13,668 before and after POM. Taking this into account, we listed the aggregated basic patient information, such as age, sex, and comorbidities, and considered to adjust the background factors. No significant difference before and after the introduction of POM was observed for any of the items.

In cases where the third assumption is discredited, wherein the incidence rate of pneumonia is reduced after the introduction of POM, an increase in the incidence rate of postoperative pneumonia relative to that before the introduction of POM may be assumed. An extension in the indication for surgery would lead to an increase in the number of geriatric patients who undergo surgery. This assumption is made given that postoperative pneumonia incident rate will increase before the introduction of POM. In such cases, the cost-effectiveness of POM may be underestimated.

### 4.4. Cost-Effectiveness Analysis of POM: Appropriateness of the Calculation of Incremental Cost-Effectiveness

The DPC data used in this study did not include increases in the costs for fee-for-service, as per the national dental fee schedule. Therefore, the dental treatment cost for pneumonia prevention can be analyzed separately. The inclusion of preventive costs in the medical DPC would have made it impossible to align the timelines for the onset of pneumonia. Thus, the appropriateness of the incremental cost calculation is ensured in this study.

### 4.5. Incremental Cost-Effectiveness Ratio and Cost-Effectiveness Analysis of POM

In the present study, we obtained the ICER by defining the prevention of postoperative pneumonia in patients with malignant tumors who received surgery as the effectiveness of POM. As a result, the ICER and reduction in medical cost were −111,927 yen/patient and −250,368,129 yen, respectively. A negative ICER indicated that the introduction of POM increased the effectiveness and lowered the cost, based on which, POM was designated as “dominant”. Although the present study assumed that POM was conducted on all patients, only 12.3% of patients with malignant tumors received POM after its introduction. Thus, the actual ICER of POM could be lower than that reported.

According to the analysis of confounding factors by Kurasawa et al. [6], the odds ratio by introduction was 0.44, and the odds ratio by gender was 2.04 for male and 4.74 by age for over 80 years old. By cancer surgery, the odds ratios were 8.95 for brain tumors, 5.49 for esophageal cancer, 5.87 for gastric cancer, and 3.85 for lung cancer. In all cases, there was a significant difference. When these factors were taken into account, all these cancer surgeries decreased in the proportion of the target group after the introduction, and the proportion of males also decreased. The average age of the patients decreased by 0.9 years. Overall, taking these results into account, we believe that the incidence of pneumonia has decreased due to the effects of the introduction of oral management, which can be calculated using the number of 275 patients.

### 4.6. Medical Cost Reduction Effect of POM

This study estimated that the introduction of POM reduced the medical costs related to postoperative pneumonia by up to 250,368,129 yen per 13,668 patients.

Dipayan et al. stated that economic evaluation of oral health care is necessary to produce the best health care and maximum benefit with minimum cost to the community based on available resources [14]. This review article states that while there are evaluations of the economic benefits of treating dental caries, periodontal disease, malocclusion, and oral cancer, there are still no evidence of economic benefits on dental fluorosis and TMJ disorders. In this context, the report does not mention the economic benefits of oral hygiene on procedures performed in the medical field. A review of the past literature does not reveal any papers of this nature, excluding the economic effects of care in other parts of the body [15].

Although it is difficult to determine whether the amount is substantial, it may be used as a reference when additional oral management-related procedures to the national health insurance scheme are introduced in other diseases in Japan.

To expand on this result globally, even in countries without universal health care coverage, a small cost for oral management would benefit patients by reducing the length of hospital stay and improving their chances of avoiding the suffering of postoperative pneumonia and its associated treatment costs.

## 5. Future Subjects

The primary limitations of this study are the variations in the details of POM (e.g., techniques and methods for oral care) between the study subjects and the rates of patients who received POM across the participating hospitals. Future studies would need to perform comparisons by incorporating data on the details of POM performed.

In addition, it is important to analyze and weigh the factors related to the differences in the hospitalization period and mortality rate. Statistically significant differences in hospitalization periods and mortality rates before and after the introduction of POM have been observed, depending on the presence of pneumonia. Nevertheless, the factors that affect these differences are not limited to the presence of postoperative pneumonia. There may be cases wherein a patient with pneumonia may have passed away from heart failure, which had developed as an intercurrent disease. From a statistical perspective, the confounding factors should be examined. A prospective study based on the results of large-scale data analysis would be useful in resolving these issues.

The fact that this is a retrospective study, which referred to DPC data, must be taken into account when conducting prospective studies in the future.

## 6. Conclusions

A multicenter study that used DPC data from approximately 25,000 postoperative cancer patients statistically found that the introduction of POM had contributed to the reduction in the incidence rate of postoperative pneumonia, the mortality rate, and the duration of hospitalization. Moreover, an analysis of the cost-effectiveness of POM indicated an ICER of approximately 110,000 yen/patient, and a reduction in medical costs through pneumonia prevention of approximately 250,000,000 yen per 13,000 patients. The findings of the current study suggest that POM effectively contributes to the reduction of medical costs.

In the comprehensive medical billing system (Diagnosis Procedure Combination) in Japan, the results of this study could only refer to the effect of reducing the hospital’s losses, but it was thought to be beneficial for patients to reduce the length of hospital stay and improve the suffering of postoperative pneumonia and the associated treatment costs, at a small cost for oral management.

In the future, we would like to attempt to conduct studies that demonstrate the usefulness of oral health care management in other countries’ health care systems.

## Figures and Tables

**Figure 1 ijerph-18-07453-f001:**
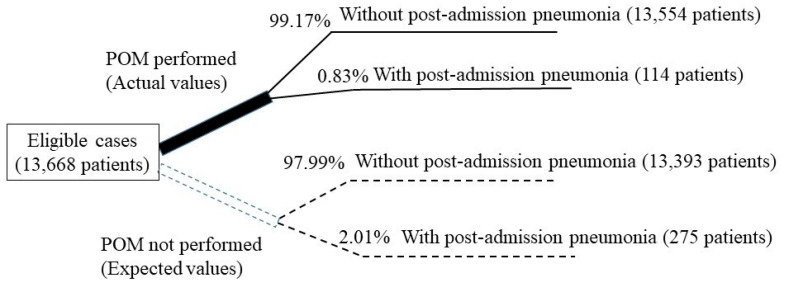
Case distribution and differentiation between the actual and expected values. Actual values are shown as a bold black line and Expected values as a dashed line. Perioperative oral management is abbreviated to POM.

**Figure 2 ijerph-18-07453-f002:**
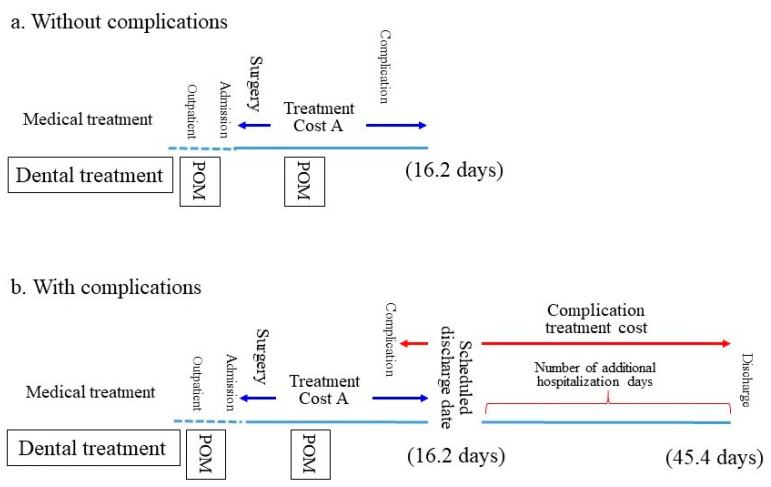
Differences in the duration of hospitalization and costs depending on the presence of complications. Numbers are the actual numbers of days of hospitalization after the introduction of perioperative oral management (POM). Dashed line shows preoperative dental treatment, blue line shows postoperative dental treatment, the range shown in dark blue represents treatment cost A (Treatment Cost A; Cancer-related treatment cost + Treatment cost for comorbidities). The range shown in red line represents treatment cost generated by complication.

**Table 1 ijerph-18-07453-t001:** Characteristics of Target group. Pre-introduction: before the introduction of perioperative oral management. Post-introduction: after the introduction of perioperative oral management.

	Pre-Introduction (*n* = 11,886)	Post-Introduction (*n* = 13,668)	*p* Value
Age (year), mean ± SD	65.5 ± 13.5	64.6 ± 13.9	<0.001
Sex (male), *n* (%)	6385 (53.7)	6577 (48.1)	<0.001
Pneumonia, *n* (%)	239 (2.0)	114 (0.8)	<0.001
Cancer type, *n* (%)			
Stomach	2060 (17.3)	2177 (15.9)	
Colon	1493 (12.5)	1657 (12.1)	
Liver and intrahepatic bile duct	1267 (10.6)	1234 (9.0)	
Rectum and anus	1118 (9.4)	1224 (8.9)	
Lung	1002 (8.4)	990 (7.2)	
Breast	965 (8.1)	2010 (14.7)	
Cervix and uterine body	917 (7.7)	1156 (8.4)	
Brain	535 (4.5)	526 (3.8)	
Esophageal	529 (4.4)	553 (4.0)	
Prostate	422 (3.5)	371 (2.7)	
Ovary and uterine appendage	258 (2.1)	283 (2.0)	
Renal	238 (2.0)	239 (1.7)	
Head and neck	220 (1.8)	277 (2.0)	
Pancreas	221 (1.8)	215 (1.5)	
Non-melanoma skin	138 (1.1)	141 (1.0)	
Thyroid gland	128 (1.0)	183 (1.3)	
Gallbladder and extrahepatic Bile duct	121 (1.0)	149 (1.0)	
Renal pelvis and ureter	94 (0.7)	129 (0.9)	
Small intestine and peritoneum	58 (0.4)	66 (0.4)	
Melanoma	51 (0.4)	41 (0.2)	
Mediastinal	31 (0.2)	30 (0.2)	
Soft tissue	16 (0.1)	15 (0.1)	
Bone	4 (0.0)	2 (0.0)	
Cornea, eye, and appendage	0 (0)	0 (0)	
Genital	0 (0)	0 (0)	
Vulva	0 (0)	0 (0)	
Vagina	0 (0)	0 (0)	
Other	0 (0)	0 (0)	

**Table 2 ijerph-18-07453-t002:** Relationship between the number of days of hospitalization and the presence of postoperative pneumonia.

		Number of Patients	Number of Days of Hospitalization (Days)		
			*Mean*	*SD*	Mann-Whitney’s *u* Test
Pre-POM introduction	Without pneumonia	11,647	17.5	16.4	4.77 × 10^−7^
	With pneumonia	239	29.8	35.8	
Post-POM introduction	Without pneumonia	13,554	16.2	16.5	4.86 × 10^−32^
	With pneumonia	114	45.4	38.7	

**Table 3 ijerph-18-07453-t003:** Relationship between treatment cost and the presence of postoperative pneumonia.

		Number of Patients	Treatment Cost (JPY)		
			*Mean*	*SD*	Mann-Whitney’s *u* Test
Pre-POM introduction	Without pneumonia	11,647	1,274,746	920,790	1.32 × 10^−2^
	With pneumonia	239	1,772,632	1,765,310	
Post-POM introduction	Without pneumonia	13,554	1,295,762	1,007,162	5.62 × 10^−30^
	With pneumonia	114	2,962,771	1,964,419	

**Table 4 ijerph-18-07453-t004:** Relationship between discharge due to death and the presence of postoperative pneumonia.

		Number of Patients	Number of Patients Discharged Due to Death	
			% (Patients)	*χ*^2^ Test
Pre-POM introduction	Without pneumonia	11,647	0.61 (71)	4.81 × 10^−6^
	With pneumonia	239	4.18 (10)	
Post-POM introduction	Without pneumonia	13,554	0.41 (56)	8.78 × 10^−7^
	With pneumonia	114	6.14 (7)	

**Table 5 ijerph-18-07453-t005:** Incidence rate of post-admission pneumonia.

	Without Pneumonia	With Pneumonia (%)	*χ*^2^ Test
Pre-POM in-troduction	11,647	239 (2.01)	7.30 × 10^−6^
Post-POM in-troduction	13,554	114 (0.83)	

## Data Availability

The data presented in this study are available on request from the corresponding author. The data are not publicly available due to privacy and ethical considerations.

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
