# Peer review of "Cost-Effectiveness Analysis of Perioperative Oral Management after Cancer Surgery and an Examination of the Reduction in Medical Costs Thereafter: A Multicenter Study"

_ijerph, 2021, doi:10.3390/ijerph18147453_

Round 1

Reviewer 1 Report

I congratulate the Authors for the their interesting study which provides new data on cost-effectiveness of POM in oncologic patients.

The study design is correct, results are clearly presented and discussed. Conclusions are scientifically sound.

Given the novelty of the approach presented for reducing the risk of developing pneumonia (systematic use of POM), it should be useful, especially for the international readers, a brief description of the protocol adopted to reduce oral cavity bacteria.

Author Response

Given the novelty of the approach presented for reducing the risk of developing pneumonia (systematic use of POM), it should be useful, especially for the international readers, a brief description of the protocol adopted to reduce oral cavity bacteria.

Answer: Thank you for your appreciative and constructive comments. Description of the protocol adopted to reduce oral cavity bacteria have been added to the Introduction section as per your suggestion.

Reviewer 2 Report

In this study, Sekiya et al evaluate the effect of national coverage and implementation of a perioperative oral management (POM) strategy in cancer patients undergoing resection and postoperative pneumonia. Using a reimbursment payment system for 8 regional hospitals in Japan, the authors evaluated the incidence of postoperative pneumonia 2 years prior to and 2 years after the initiation of POM coverage by the nation. Patients hospitalized for pneumonia prior to surgery or those not undergoing surgery for solid tumors were excluded. The overall concept of POM and its potential to reduce postoperative pneumonia is of great significance in public health; however, several methodological issues limit the validity of the study. A significant issue with the assumptions is that 1) likely some form of POM was already being practiced prior to the nationwide implementation? Defining a specific data were POM was inplemented (April 2012) is necessary for analysis; however, not alway realistic as initiation of such a program is gradual. The second assumption is unclear how the authors came with 12.3% proportion of POM use prior to data analysis. This needs further explanation. As for the third assumption, in any time series analysis, ajdustment for multiple factors are necessary as there are unbalanced confounders in both groups (age, type of surgery, sex, etc). This is particularly true when evaluating table 1. Apart from differences in age and sex, there were clear reductions in digestive, thoracic and neurologic surgery prior to implementation while post-implementation, breast surgery was more prevalent. The different surgery types likely has an effect on incidence of postoperative pneumonia and should be adjusted by multivariable analysis. As such, estimated the number of patients who would have developed postoperative pneumonia requires adjustment and likely would be less than 275 patients. As the subsequent cost analysis is based on these values, the cost-effectiveness and ICER would need to be revised as well. 

Minor comments: 
Methods: 2.3 Cost-effectiveness analysis of POM. The first line of this paragraph should state that while multiple methods exist to evaluate cost in healthcare, the authors selected cost-effectiveness. 

The authors should clearly state what POM entails. Is it dental evaluation and dental treatment? Does it differ per patient or is standardized?

Why was the cost associated with pneumonia increased from 2012-2014 compared to the prior years?

Discussion: 4.1 - The authors should reference the classification system used for pneumonia (acute, bacterial, VAP, aspiration) as all have overlap and are not mutually exclusive. 

The authors should provide a table with regression analysis of covariates to evaluate the effect on postoperative pneumonia (age, sex, surgery type, tumor stage, comorbidity, etc).

Author Response

In this study, Sekiya et al evaluate the effect of national coverage and implementation of a perioperative oral management (POM) strategy in cancer patients undergoing resection and postoperative pneumonia. Using a reimbursement payment system for 8 regional hospitals in Japan, the authors evaluated the incidence of postoperative pneumonia 2 years prior to and 2 years after the initiation of POM coverage by the nation. Patients hospitalized for pneumonia prior to surgery or those not undergoing surgery for solid tumors were excluded. The overall concept of POM and its potential to reduce postoperative pneumonia is of great significance in public health; however, several methodological issues limit the validity of the study.

A significant issue with the assumptions is that 1) likely some form of POM was already being practiced prior to the nationwide implementation? Defining a specific data were POM was implemented (April 2012) is necessary for analysis; however, not always realistic as initiation of such a program is gradual.

Answer: Thank you for the suggestion. We have added a structural description regarding your advice in Subjects and methods as follows, “The analyses were limited to cancer surgery because of the initial introduction of the patients into the Japanese social healthcare system and the lack of perioperative oral management for cancer patients prior to the addition to the fee schedule, which simplified the statistical analyses.”

The second assumption is unclear how the authors came with 12.3% proportion of POM use prior to data analysis. This needs further explanation.

Answer: Thank you for the comment. We have added a description regarding your advice in Subjects and methods as follows, “Since the medical DPC data and dental data are sent to the data center in an unlinked and anonymized form, the dental intervention rate is discussed as unknown in the paper by Kurasawa et al.6). In this study, we calculated a dental intervention rate of 12.3% by manual data matching at the Yokohama City Minato Red Cross Hospital, where the data remained in the hospital in a linkable form. This was used as the second assumption.”

As for the third assumption, in any time series analysis, adjustment for multiple factors are necessary as there are unbalanced confounders in both groups (age, type of surgery, sex, etc). This is particularly true when evaluating table 1. Apart from differences in age and sex, there were clear reductions in digestive, thoracic and neurologic surgery prior to implementation while post-implementation, breast surgery was more prevalent. The different surgery types likely has an effect on incidence of postoperative pneumonia and should be adjusted by multivariable analysis. As such, estimated the number of patients who would have developed postoperative pneumonia requires adjustment and likely would be less than 275 patients. As the subsequent cost analysis is based on these values, the cost-effectiveness and ICER would need to be revised as well.

Answer: Thank you for the comment. We have added a description regarding your advice in Introduction as follows, “This data is based on the same database as the paper by Kurasawa et al. published in 20206). Kurasawa et al. belongs to the same study group and is in charge of the statistics. For the data of about 25,000 cancer patients who underwent surgery, the incidence of pneumonia before and after the introduction of insurance was discussed using multivariate analysis considering confounding factors, and it was concluded that the introduction of insurance significantly reduced the incidence of pneumonia6). and Subjects and methods as follows, “However, introducing of insurance system is not the only factor involved in the re-duction of pneumonia; Kurasawa et al. conducted a multivariate logistic analysis using the same database to investigate the involvement of confounding factors6). Other con-founders included older than 70 years, male, brain tumor, esophageal cancer, gastric cancer, and lung cancer surgery (P<0.01). Of these, in the post-implementation group, the mean age decreased and the proportion of males decreased, but the proportions accounted for by brain tumor, esophageal cancer, gastric cancer, and lung cancer surgery decreased, and it was unlikely that these factors had any effect on reducing the incidence of pneumonia.” And we have added a description in Discussion4.2. and 4.5 as follows, “Using the same target data as in this study, Kurasawa et al. 6) conducted a multivariate analysis to investigate the relationship with confounding factors, and showed the effectiveness of the introduction of oral management in preventing pneumonia after cancer surgery.”, and “According to the analysis of confounding factors by Kurasawa et al, the odds ratio by introduction was 0.44, and the odds ratio by gender was 2.04 for male and 4.74 by age for over 80 years old. By cancer surgery, the odds ratios were 8.95 for brain tumors, 5.49 for esophageal cancer, 5.87 for gastric cancer, and 3.85 for lung cancer. In all cases, there was a significant difference. When these factors were taken into account, all these cancer surgeries decreased in the proportion of the target group after the introduction, and the proportion of males also decreased. The average age of the patients decreased by 0.9 years. Overall, taking these results into account, we believe that the incidence of pneumonia has decreased due to the effects of the introduction of oral management, which can be calculated using the number of 275 patients.”

I have attached the Kurasawa’s paper, which uses the common database. Please refer to it.

Thank you very much for your kind advice.

Reviewer 3 Report

an interesting paper with some points to fix.

  1. a literature review is missing and references listed are very few and often dated.
  2. Improve the list through https://scholar.google.it/scholar?as_ylo=2017&q=medical+cost+effectiveness+survey&hl=it&as_sdt=0,5 or other sources (Sopus, ISI WoS; researchgate, etc.)
  3. introduce a brief§ after the introduction with a short review of the literature, explaining why your study is innovative
  4. back the methodology with quotations, explaining why the method that you use is consistent with the literature; describe better cost/benefit analysis in healthcare linking it to previous studies
  5. improve discussion section 5, extending future research streams (other pathologies to be analyzed with a similar method? extension of the study to other countries? ...)
  6. improve the conclusion, stressing even more why the study is interesting even outside Japan, to which extent the results can be generalized elsewhere, how the healthcare benefits and savings can directly accrue to the patients ...

Author Response

an interesting paper with some points to fix.

a literature review is missing and references listed are very few and often dated.

Improve the list through other sources (Sopus, ISI WoS; researchgate, etc.)

introduce a brief§ after the introduction with a short review of the literature, explaining why your study is innovative back the methodology with quotations, explaining why the method that you use is consistent with the literature; describe better cost/benefit analysis in healthcare linking it to previous studies

improve discussion section 5, extending future research streams (other pathologies to be analyzed with a similar method? extension of the study to other countries? ...) improve the conclusion, stressing even more why the study is interesting even outside Japan, to which extent the results can be generalized elsewhere, how the healthcare benefits and savings can directly accrue to the patients.

Answer: Thank you for your appreciative and constructive comments. We have surveyed the literature related to cost-effectiveness analysis and medical costs reduction, and cited it, adding its content to the discussion. And as you mentioned, we have improved section 5 of the discussion and included it in the manuscript. Thank you very much for your help again.